# Automated Emotion Identification Using Fourier–Bessel Domain-Based Entropies

**DOI:** 10.3390/e24101322

**Published:** 2022-09-20

**Authors:** Aditya Nalwaya, Kritiprasanna Das, Ram Bilas Pachori

**Affiliations:** Department of Electrical Engineering, Indian Institute of Technology Indore, Indore 453552, India

**Keywords:** Fourier–Bessel series expansion, spectral entropy, ECG, EEG, FBSE-EWT

## Abstract

Human dependence on computers is increasing day by day; thus, human interaction with computers must be more dynamic and contextual rather than static or generalized. The development of such devices requires knowledge of the emotional state of the user interacting with it; for this purpose, an emotion recognition system is required. Physiological signals, specifically, electrocardiogram (ECG) and electroencephalogram (EEG), were studied here for the purpose of emotion recognition. This paper proposes novel entropy-based features in the Fourier–Bessel domain instead of the Fourier domain, where frequency resolution is twice that of the latter. Further, to represent such non-stationary signals, the Fourier–Bessel series expansion (FBSE) is used, which has non-stationary basis functions, making it more suitable than the Fourier representation. EEG and ECG signals are decomposed into narrow-band modes using FBSE-based empirical wavelet transform (FBSE-EWT). The proposed entropies of each mode are computed to form the feature vector, which are further used to develop machine learning models. The proposed emotion detection algorithm is evaluated using publicly available DREAMER dataset. K-nearest neighbors (KNN) classifier provides accuracies of 97.84%, 97.91%, and 97.86% for arousal, valence, and dominance classes, respectively. Finally, this paper concludes that the obtained entropy features are suitable for emotion recognition from given physiological signals.

## 1. Introduction

Nowadays, humans are becoming more and more dependent on computers for their various day-to-day tasks. Thus, the need for making computers more user-friendly and engaging is becoming more and more common rather than just being logically or computationally efficient [1]. To make computers more user-friendly, it must recognize the emotion of the user interacting with it, as emotion is the most basic component that is responsible for human attention, action, or behavior in a particular situation [2]; therefore, for the applications related to human–computer interaction (HCI), an emotion recognition system is very useful [3]. Further, such HCI system can be very useful in areas related to mental health care [4], smart entertainment [5], and an assistive system for a physically disabled person [6].

Emotion recognition systems generally have two different approaches for recognizing emotion, i.e., explicit approach, which includes expression on the face, speech, etc., that are visible and can be easily captured for analysis; however, the problem with such an approach is that the signals obtained in this approach are not very reliable because the subject under consideration can hide his/her emotion, so these might not show the actual emotional state of the subject. An alternative way of recognizing emotion is an implicit approach where physiological signals, such as electroencephalogram (EEG), electrocardiogram (ECG), galvanic skin response (GSR), etc., are being captured and analyzed. Such signals are not visible and are generated inside the body by the autonomous nervous system (ANS). Any change in the emotional state of a person is reflected in the respiration rates, body temperature, heart rate, and other physiological signals, which are not under the control of the subject [7].

The brain is a central command center. It reacts to any external stimuli by firing neurons inside the brain, causing changes in ANS activity, due to which, the routine activity of the heart and other peripheral organs varies. In addition, change in the emotional state affects heart activity [8]. EEG and ECG signals are the biopotential that reflect the activities of the brain and heart, respectively. Thus, in this study, to recognize the emotion of the subject, physiological signals namely, EEG and ECG are considered. Although in the literature, some other physiological signals are also used to determine the emotional state of a person, such as changes in the conductivity of the skin, which is measured using GSR [9,10,11]. In [12], the authors have extracted time–frequency domain features using fractional Fourier transform (FrFT); the most relevant features are selected using the Wilcoxon method, which is then given as input to a support vector machine (SVM) classifier for determining the emotional state of the person. Similarly, the respiration rate also helps in determining the emotional state of the person. In [13], authors have used deep learning (DL)-based feature extraction method and logistic regression, determining different emotional states from the obtained features.

In recent years, several emotion recognition frameworks have been established using different physiological signals. The authors in [14] have proposed a methodology that uses ECG signal features derived from different intrinsic mode functions (IMFs) for emotion classification. Using bivariate empirical mode decomposition (BEMD), IMFs are obtained. The instantaneous frequency and local oscillation feature are computed, which are given as input to the linear discriminant classifier to discriminate different emotional states. A method is proposed to determine negative emotion through a single channel ECG signal [15]. Different features, extracted from the ECG signals, are linear features, i.e., mean of RR interval. Nonlinear features consist of bispectral analysis, power content in different frequency bands, time–domain features contains different statistical parameters of ECG signal, and time–frequency domain features were obtained using wavelet-based signals decomposition. Based on these features, different emotions were detected using several machine learning classifiers. The authors of [16] have extracted four different features from the ECG signal, i.e., heart rate variability (HRV), with-in beat (WIB), frequency spectrum, and signal decomposition-based features. To evaluate the performance of the obtained features, ensemble classifiers are used. In [17], the authors have extracted ECG signal features at different time scale using wavelet scattering, and features obtained are given as an input to various classifiers for evaluating their performance.

Similarly, different emotion recognition techniques have been proposed based on EEG signals. Anuragi et al. [18] have proposed EEG based emotion detection framework. Using the Fourier–Bessel series expansion (FBSE)-based empirical wavelet transform (EWT) (FBSE-EWT) method, the EEG signals are decomposed into four sub-band signals from which features such as energy and entropy are extracted. The features are selected using neighborhood component analysis (NCA), using different classifiers, such as k-nearest neighborhood (KNN), ensemble bagged tree, and artificial neural network (ANN), emotion class is identified. Sharma et al. [19] have used discrete wavelet transform (DWT) for EEG signal decomposition. Third-order cumulant nonlinear features are extracted from each sub-band, and using swarm optimization features, dimensionality is reduced. To classify the emotional states, a long short-term memory-based technique is used. Bajaj et al. [20] have used multiwavelet transform to decompose EEG signals into narrow sub-signals. Using Euclidean distance, features are extracted that are computed from a 3-D phase space diagram. Multiclass least squares support vector machines (MC-LS-SVM) classifier is used for identifying class of emotion; however, the authors in [21] have computed features, namely entropy and ratio of the norms-based measure, are computed from the sub-signals that are obtained after decomposing EEG signal using multiwavelet decomposition. The feature matrix is given as input to MC-LS-SVM for determining the emotional state. In [22], features related to changes in the spectral power of EEG signal are extracted using a short-time Fourier transform (STFT). The best 55 features among 60 features are selected using F-score. A feature vector is formed from the selected features which are given as an input to the SVM classifier for determining the emotional state of the subject. Liu et al. [23] have extracted fractal dimension and higher-order crossing (HOC) as features via the sliding window approach from the given EEG signal; based on these features, an emotion class is determined using an SVM classifier. In [24], the authors have decomposed EEG signals in different sub-bands using DWT. From the obtained wavelet coefficient features, entropy and energy of the coefficients are calculated. Finally, using SVM and KNN classifiers, emotional states are obtained.The authors in [25] have used a multi-channel signal processing technique called multivariate synchrosqueezing transform (MSST) for representing EEG signals in time–frequency domain. Independent component analysis (ICA) is used to reduce dimensionality of the obtained high-dimensional feature matrix. In addition, its performance is compared with non-negative matrix factorization (NMF), which is an alternative feature selection method. The reduced feature matrix is passed to different classifiers such as SVM, KNN, and ANN to discriminate different emotions. Gupta et al. used flexible analytic wavelet transform for decomposing EEG signals into narrow-band sub-bands [26]. From each sub-band, information potential (IP) is computed using Reyni’s quadratic entropy, and using moving average filter the extracted features are smoothed. To determine the different emotion classes, a random forest classifier is used. Ullah et al. segmented the large duration EEG signal into smaller epoch [27] from each segment features; specifically, Fisher information ratio, entropy, statistical parameters, and Petrosian fractal dimension, are computed. The above feature vector is passed to a sparse discriminative ensemble learning (SDEL) classifier to identify the emotion class. Bhattacharyya et al. decomposed EEG signals into different modes using FBSE-EWT [28]. From each mode of KNN entropy, multiscale multivariate Hilbert marginal spectrum (MHMS), and spectral Shannon entropy, features are calculated. Features are smoothed and fed to a classifier called sparse autoencoder-based random forest (ARF) for determining the class of human emotion. Nilima et al. used the empirical mode decomposition (EMD) method for decomposing EEG signal [29]. The second-order difference plot (SODP) features are calculated from each IMF. A two-hidden layer multilayer perceptron is used for multi class classification and SVM is used for binary classification. Nalwaya et al. [30], have used tunable Q-factor wavelet transform (TQWT) to separate various rhythms of EEG. Different statistical and information potential features are computed for each rhythm, which are then fed to SVM cubic classifier for emotion identification.

Further, some of the studies use both ECG and EEG signals. In [31], the authors have recorded EEG and ECG signals while the subject is exposed to immersive virtual environments to elicit emotion. From ECG signal, time–domain features, frequency domain features, and non-linear features are calculated. Whereas from EEG signal band power and mean phase connectivity features are calculated. Both EEG and ECG features are combined to form a single feature matrix whose dimensionality is then reduced using principal component analysis (PCA). Then, the reduced matrix is passed to an SVM classifier to determine the emotional state of the subject.

The literature review of previous studies on human emotion identification shows emerging research trends in finding appropriate signal decomposition techniques, distinctive features, and machine learning algorithms to classify emotional states. Most of the studies carried out previously used a single modality, i.e., either EEG, ECG, or other peripheral physiological signals, and very few studies have been conducted using multiple modalities. Despite this, there is still scope for improvement in the classification accuracy of the earlier proposed methods.

In this article, an automated emotion identification framework using EEG and ECG signals is developed. Furthermore, the Fourier–Bessel (FB) domain has been explored instead of working in traditional Fourier domain, due to various advantages associated with latter one. FBSE uses Bessel functions as basis functions whose amplitude decay with time and are damped in nature, which make them more suitable for non-stationary signal analysis. FBSE spectrum has twice the resolution as compared to Fourier domain spectral representation. Thus, looking at such advantages, FBSE-EWT is used instead of EWT for extracting different modes from the given signal. New FB-based spectral entropy, such as Shannon spectral entropy (SSE), log energy entropy (LEE), and Wiener entropy (WE), have been proposed, which are used as features from the obtained modes. Then, smoothing of the feature values is performed by applying moving average over obtained features values, which is then given as input to SVM and KNN classifiers for emotional class identification. The block diagram of the proposed methodology is shown in Figure 1.

The rest of the article is organized as follows: In Section 2, the material and methodology are discussed in detail. In Section 3, results obtained after applying the proposed methodology are presented. Section 4 presents the performance of the proposed method with the help of the results obtained, compares it with other exiting methodologies, and highlights some of its limitations. Finally, Section 5 concludes the article.

## 2. Materials and Methods

The emotion recognition framework presented in this article consists of preprocessing, signal processing, feature extraction, feature smoothing, and classification steps.

### 2.1. Dataset Description

A publicly available DREAMER dataset is used to evaluate the proposed methodology. It contains raw EEG and ECG signals, which are recorded from 23 healthy participants while the subject is watching audio and video clips. Emotions are quantified in terms of three different scales: arousal, valence, and dominance [32]. Each participant was shown 18 different clips of different durations with a mean of 199 s. The EPOC system by Emotive was used, which contains 16 gold-plated dry electrodes placed in accordance with the international 10–20 system EEG, and were recorded from, i.e., AF3, AF4, F3, F4, F7, F8, FC5, FC6, T7, T8, P7, P8, O1, and O2; two reference electrodes, M1 and M2 were placed over mastoid, as described in [32]. To obtain the ECG signals, electrodes were placed in two vector directions, i.e., right arm, left leg (RA-LL) vector and left arm, left leg (LA-LL) vector. ECG was recorded using the SHIMMER ECG sensor. Both EEG and ECG signals were recorded at different sampling rates, i.e., 128 Hz, and 256 Hz, respectively. The sample EEG and ECG signals obtained from the dataset are shown in Figure 2 for different classes of emotion (high arousal (HA) or low arousal (LA), high dominance (HD) or low dominance (LD), high valence (HV) or low valence (LV)). The dataset also contains information relating to the rating of each video on a scale of 1 to 5. Each participant rated the video as per his/her level of emotion elicited in three different dimensions, i.e., valence (pleasantness level), arousal (excitement level), and dominance (level of control). Rating between 1–5 was labeled as ‘0’ (low) or ‘1’ (high) for each dimension, with 3 as the threshold, i.e., if a participant rates a video between 1 to 3, it will be considered as low or ‘0’ and a value above 3 is considered as high or ‘1’ [33,34].

### 2.2. Preprocessing

At this stage, typically, noise and artifacts from the raw signal are removed with the help of filters and denoising algorithms. As in [32], the analysis of only the last 60 s of signal was performed. Further, signals were segmented into small epochs of 1 s, which were then used for further processing. The mean value was subtracted from each epoch.

### 2.3. FBSE-EWT

Analyzing non-stationary signals such as EEG and ECG is difficult as the signal is time-varying in nature and its properties also change continuously. In order to understand the properties of such signals, decomposing them into narrow-band simpler components can help to make it easier to understand; therefore, the preprocessed signal was given to a signal decomposition algorithm, which will decompose the input EEG and ECG signal into various modes.

FBSE-EWT is an improved version of the EWT, which has adaptive basis functions derived from the signals. FBSE-EWT has been used for many biomedical signal processing applications, such as epileptic seizure detection [35,36], valvular heart disease diagnosis [37], and posterior myocardial infarction detection [38]. FBSE-EWT technique flow is shown in Figure 3 and its step by step working is as follows:

1.FBSE spectrum: FBSE has twice the frequency resolution as compared to the Fourier domain. FBSE spectrum of a signal s(n) of length *S* samples can be obtained using zero-order Bessel functions. The magnitude of the FB coefficients K(i) can be computed mathematically as follows [39,40,41,42]:
(1)$$K\left(i\right)=\frac{2}{S^{2}{\left[J_{1}\left(\beta_{i}\right)\right]}^{2}}\sum_{n=0}^{S-1}ns\left(n\right)J_{0}\left(\frac{\beta_{i}n}{S}\right)$$
where $J_{0}\left(\cdot \right)$ is the zero-order and $J_{1}\left(\cdot \right)$ is the first-order Bessel functions. Positive roots of zeroth order Bessel function are denoted by $\beta_{i}$, which are arranged in ascending order, where i=1,2,3,⋯,S. A one to one relation between order *i* and continuous frequency is given by [39,42],
(2)$$\beta_{i}\approx \frac{2\pi f_{i}S}{f_{s}}$$
where $\beta_{i}\approx \beta_{i-1}+\pi \approx i\pi$, $f_{s}$ denotes the sampling frequency. Here, $f_{i}$ is the continuous frequency, corresponding to the $i^{\mathrm{th}}$ order and is expressed as [39],
(3)$$i\approx \frac{2f_{i}S}{f_{s}}$$For covering whole bandwidth of s(n), *i* must vary from 1 to *S*. The FBSE spectrum is the plot of magnitude of the FB coefficient K(i) versus frequency $f_{i}$.2.Scale-space based boundary detection [39,42,43]: For FBSE spectrum, scale-space representation can be obtained by convolving the signal with a kernel of Gaussian type, which is expressed as follows:
(4)$$\vartheta \left(i,q\right)=\sum_{n=-N}^{N}K\left(i-n\right)g\left(n;q\right)$$
where $g\left(n;q\right)=\frac{1}{\sqrt{2\pi q}}e^{\frac{-n^{2}}{2q}}$. Here, $N=W\sqrt{q}+1$ with 3≤W≤6 and *q* is the scale parameter. As the scale-space parameter, i.e., $\varepsilon =\sqrt{\frac{q}{q_{0}}}$, $\varepsilon =1,2,\cdots ,\varepsilon_{\max}$, increases, the number of minima decreases and no new minima will appear in the scale space plan. The FBSE spectrum is segmented using the boundary detection technique. The FBSE spectrum ranges from 0 to π and the FBSE spectrum segments are denoted as $\left[0,\omega_{1}\right]$, $[\omega_{1},\omega_{2}],\cdots ,$ and $\left[\omega_{i-1},\pi \right]$, where $\omega_{1},\omega_{2},\cdots ,\omega_{i-1}$ are boundaries. Typically boundaries are defined between two local minima, which are obtained by the two curves in the scale–space plane, whose length is greater than the threshold, which is obtained by using Otsu method [44].3.EWT filter bank design: After obtaining the filter boundaries, based on these parameters, empirical scaling and wavelet functions are adjusted and different band-pass filters are designed. Wavelets scaling ($\Phi_{j}\left(\omega \right))$ and wavelet functions ($\nu_{j}\left(\omega \right)$) were constructed, and the mathematical expressions are given by [45],
(5)$$\Phi_{j}\left(\omega \right)=\left\{\begin{array}{cc} 1 & \;\mathrm{if}\;\left|\omega \right|\leq \left(1-\tau \right)\omega_{j}\\ \cos [\frac{\pi \gamma (\tau ,\omega_{j})}{2}] & \;\mathrm{if}\;\left(1-\tau \right)\omega_{j}\leq \left|\omega \right|\leq \left(1+\tau \right)\omega_{j}\\ 0 & \mathrm{otherwise} \end{array}\right.$$
(6)$$\nu_{j}\left(\omega \right)=\left\{\begin{array}{cc} 1 & \;\mathrm{if}\;\left(1+\tau \right)\omega_{j}\leq \left|\omega \right|\leq \left(1-\tau \right)\omega_{j+1}\\ \cos [\frac{\pi \gamma (\tau ,\omega_{j+1})}{2}] & \;\mathrm{if}\;\left(1-\tau \right)\omega_{j+1}\leq \left|\omega \right|\leq \left(1+\tau \right)\omega_{j+1}\\ \sin [\frac{\pi \gamma (\tau ,\omega_{j})}{2}] & \;\mathrm{if}\;\left(1-\tau \right)\omega_{j}\leq \left|\omega \right|\leq \left(1+\tau \right)\omega_{j}\\ 0 & \mathrm{otherwise} \end{array}\right.$$
where, $\gamma \left(\tau ,\omega_{j}\right)=\delta [\frac{\left|\omega \right|-\left(1-\tau \right)\omega_{j}}{2\tau \omega_{j}}]$. To obtain the tight frames parameter, δ and τ are defined as
(7)$$\delta \left(x\right)=\left\{\begin{array}{cc} 0 & \;\mathrm{if}\;x\leq 0\\ \; & \mathrm{and}\;\delta (x)+\delta (1-x)=1\;\;\;\forall \;x\;\in [0,\;1]\\ 1 & \;\mathrm{if}\;x\geq 1 \end{array}\right.$$
(8)$$\tau <\min_{j} [\frac{\omega_{j+1}-\omega_{j}}{\omega_{j+1}+\omega_{j}}]$$4.Filtering: Expression for detailed coefficients from the EWT filter bank for the analyzed signal a(m) is given by [45],
(9)$$D_{a,\nu }\left(j,n\right)=\int a\left(m\right)\bar{\nu_{j}\left(m-n\right)}dm$$The approximate coefficients from the EWT filter bank are computed by [45],
(10)$$A_{a,\Phi }\left(0,n\right)=\int a\left(m\right)\bar{\Phi_{1}\left(m-n\right)}dm$$The $j^{\mathrm{th}}$ empirical mode is obtained by convolving the wavelet function with the detail coefficients, where j=1,2,⋯,M are different empirical modes. Original signal can be reconstructed by adding all *M* reconstructed modes and one low-frequency component; mathematically, both are expressed as below [45]
(11)$$r_{j}\left(n\right)=D_{a,\nu }\left(j,n\right)★\nu_{j}\left(n\right)$$
(12)$$r_{0}\left(n\right)=A_{a,\Phi }\left(0,n\right)★\Phi_{1}\left(n\right)$$
where ★ denotes the convolution operation.

### 2.4. Feature Extraction

In order to understand and quantify information associated with any dynamically changing phenomenon, entropy is the most widely used parameter [46]. Thus, in order to understand the nonstationary behavior of physiological signals considered in this study, entropies, as a feature set, are considered. Some of the advantages of FBSE representation are that Bessel functions decay with time, and so provide more suitable representation of the nonstationary signal, and that FBSE has a frequency resolution twice that of the Fourier spectrum. In addition, there are many previous studies on emotion recognition where entropies have been used [26,28,42,47]. SSE, WE, and LEE are defined as follows:

#### 2.4.1. SSE

Uniformity in the distribution of signal energy can be measured using spectral entropy. Entropy measures the uncertainty, which has been derived from the Shannon’s expression. The SSE is defined based on the energy spectrum $E_{i}$ obtained using FBSE. The energy corresponding to the $i^{\mathrm{th}}$ order FB coefficient K(i) is defined mathematically as [48]
(13)$$E_{i}=\frac{K{\left(i\right)}^{2}S^{2}{\left[J_{1}\left(\beta_{i}\right)\right]}^{2}}{2}$$

The SSE is expressed mathematically as [49]
(14)$$H_{\mathrm{SSE}}=-\sum_{i=1}^{S}P\left(i\right)\;\log_{2}\left(P(i)\right)$$
where *S* is the total number of FBSE coefficients. P(i) is the normalized energy distribution over $i^{\mathrm{th}}$ order is mathematically defined as,
(15)$$P\left(i\right)=\frac{E_{i}}{\sum_{i=1}^{S}E_{i}}$$

#### 2.4.2. WE

It is another measure of flatness in the distribution of signal spectral power. The WE is also called the measure of spectral flatness. It is calculated by taking the ratio of geometric mean to arithmetic mean of the energy spectrum ($E_{i}$) of the FB coefficient for order *i* and is expressed as [50],
(16)$$H_{\mathrm{WE}}=S\frac{\sqrt[{S}]{\prod_{i=1}^{S}E_{i}}}{\sum_{i=1}^{S}E_{i}}$$
WE is a unitless quantity; its output is purely a numerical value ranging from 0 to 1, where 1 represents a uniform (flat) spectrum and 0 indicates a pure tone.

#### 2.4.3. LEE

Another variant of information measurement using LEE is defined as logarithm of P(i) in the FB domain and it is given by [51]
(17)$$H_{\mathrm{LE}}=-\sum_{i=1}^{S}\;\log_{2}\left(P(i)\right)$$

### 2.5. Feature Smoothing

The brain is the control center of all human activities and internal functioning. The typical EEG signal obtained is thus a combination of various brain activity and other noise, either related to the environment or body [52]. Rapid fluctuations in feature value may arise from these noises. Human emotion changes gradually [26]; in order to reduce the effect of such variation on the emotional state-related feature values, a moving average filter with a window size of 3 samples is utilized. The effect of the moving average filter on the raw feature value of the first channel’s first epoch can be seen in Figure 4.

### 2.6. Classifiers

Feature vectors extracted from the EEG signals are used for the classification of different emotions. Based on the feature vector, the classifier will discriminate the data into high and low dimensions of emotion, i.e., either HV or LV, HA or LA, and HD or LD. In this study, SVM with the cubic kernel and KNN are used independently for the purpose of classification.

SVM is a supervised machine learning algorithm that classifies data by first learning from labeled data belonging to different classes. The main principle behind the classifier working is finding decision boundaries that are formed by a hyperplane, and it helps in separating data into two separate classes. The hyperplane is constructed by training the classifier with sample data. The optimum location of the hyperplane or decision boundary depends on the support vectors, which are the points nearest to the decision boundary [53]. SVM iteratively optimizes the location of hyperplane in order to maximize the margin. A margin is a total separation between two classes. SVM can be linear or nonlinear classifier depending on kernels used. Generally, in the case of a linear SVM straight line, flat plane, or an N-dimensional hyperplane are the simplest way of separating data into two groups, but there are certain situations where nonlinear boundary separates the data more efficiently. In case of a nonlinear classifier, different kernels can be polynomial, hyperbolic tangent function, or Gaussian radial basis function. In this work, an SVM classifier with a cubic kernel is used. Generalized mathematical expression for defining the hyperplane of the SVM classifier can be expressed as follows [54]:(18)$$f\left(x\right)=\mathrm{sign} [\sum_{i=1}^{R}b_{i}\;f_{i}\;\;K\left(x,x_{i}\right)+c]$$
where $b_{i}$ is a positive real constant, *R* is total number of observations, *c* is a real constant, $K(x,x_{i})$ is a kernel or feature space, $x_{i}$ input vector, and output vector is denoted by $f_{i}$. For a linear feature space $K\left(x,x_{i}\right)=x_{i}^{T}x$, for polynomial SVM of any order *p*, $K\left(x,x_{i}\right)={\left(x_{i}^{T}x+1\right)}^{p}$ defines the feature space, i.e., (p=2) for quadratic polynomial and (p=3) for cubic polynomial.

The KNN is a non-parametric, supervised machine learning algorithm used for data classification and regression [55]. It categorizes data points into different groups based on the distance from some of the nearest neighbors. For classifying any particular training data, the KNN algorithm follows these steps:Step 1:Compute distance between sample data and other sample using anyone of the distance metrics such as Euclidean, Mahalanobis, or Minkowski distance.Step 2:Rearrange the distant metric obtained from the first step in ascending order and top *k* values are considered with distance from current sample is minimum.Step 3:Class is assigned to the sample data depending on the maximum number of nearest neighbors class.

## 3. Results

In order to evaluate the performance of our proposed methodology, a publicly available DREAMER dataset is used [32]. In this section, the results obtained after applying the proposed emotion identification method over the DREAMER dataset is discussed in detail. The raw EEG data are segmented into an epoch of one second. An EEG epoch length is 128 samples and the ECG length is 256 samples. These epochs of the EEG signal are decomposed into four modes using FBSE-EWT, as shown in Figure 5, and the corresponding filter bank is shown in Figure 6. The decomposition of EEG and ECG signal epochs gives a minimum of four number of modes. So, we set the required number of modes to four to obtain a uniform number of features across different observations. From each mode, three features are extracted. The EEG data consist of 14 channels; therefore, the dimension of the feature vector is 168 (=14×3×4). Similarly, from the two channels of the ECG, we obtain a 24-element feature vector. The number of trials for each participant is 18 and there are a total of 23 participants, which gives us a total of 24,840 (=60×18×23) observations or epochs. The final dimensions of the feature matrices are 24,840 ×168 and 24,840 ×24 for EEG and ECG, respectively. For the multimodal case, both the feature from EEG and ECG are combined, where we obtain a feature matrix with the dimension of 24,840 ×192. This feature matrix is classified using two classifiers: SVM and KNN. For KNN (k=1), Euclidean distance is used as a distance metric; for SVM, cubic kernels are used.

Performance was evaluated using the classifier learner application present in MATLAB 2022a. The system used for computing has an Intel i7 CPU with 8 GB of RAM. For three different dimensions of emotion, i.e., arousal, dominance, and valence, three different binary classifications are performed. Thus, each classifier groups data into either high or low arousal, high or low dominance, and high or low valence classes. Low and high class is decided based on the rating given by the participant on a scale of 1 to 5, with 3 as threshold, i.e., rating ≤ 3 is consider low and all above it is considered as high [33,34], due to which, unbalanced classes are obtained for some participants [32]. Further, the performance of the proposed method is evaluated using the first features obtained from EEG signals, then ECG signals, and then using the combined features.

Features obtained from EEG and ECG signals of different subjects are combined together and the classifiers are trained and tested subject independently. Both SVM and KNN are evaluated independently using five-fold cross-validation, where different signal feature observations are placed randomly independent to the subject. For the EEG feature and arousal, the dimension accuracy obtained for SVM is 81.53% and for KNN it is 97.50%. Similarly, performance was evaluated for the case of dominance and valence, which are summarized in Table 1, Table 2 and Table 3. The tables consist of classification results obtained by using different modalities, i.e., EEG, ECG, and the combined multimodal approach, and the best results obtained are highlighted using bold fonts. The confusion matrices for three different emotion dimensions classifications based on EEG, ECG, and multimodal signals are shown in Figure 7, Figure 8 and Figure 9.

Accuracy parameters, such as sensitivity, specificity, precision, and F1 score, are shown in Table 1, Table 2 and Table 3. From these tables, it may be observed that the accuracy obtained from EEG and multimodal are almost similar; however, among the two, multimodal is still the winner as it can be seen that all other parameters of classifier reliability check are having sightly better result than the unimodal EEG signal. Following mathematical expressions have been used for calculating parameters namely sensitivity, specificity, precision, and F1 score [53]:(19)$$\mathrm{Sensitivity}=\frac{\mathrm{TP}}{\mathrm{TP}\;+\;\mathrm{FN}}$$
(20)$$\mathrm{Specificity}=\frac{\mathrm{TN}}{\mathrm{FP}\;+\;\mathrm{TN}}$$
(21)$$\mathrm{Precision}=\frac{\mathrm{TP}}{\mathrm{TP}\;+\;\mathrm{FP}}$$
(22)$$F1\;\mathrm{Score}=\frac{2\mathrm{TP}}{2\mathrm{TP}\;+\;\mathrm{FP}\;+\;\mathrm{FN}}$$
where TP is the true positive, TN is the true negative, FP is the false positive, and FN is the false negative.

## 4. Discussion

Although, in recent years, significant research has been carried out on emotion recognition related topics, still, it is challenging due to the fuzziness in distinction among different emotions. The study presented in this article demonstrated an approach for developing an automated human emotion identification that is able to categorize the three dimensions of emotion, i.e., arousal, valence, and dominance, into high or low classes of a particular dimension. In order to retrieve emotion-related information from the physiological signals, the FBSE-EWT signal decomposition method is used. Then, various FB-based entropies, such as SSE, WE, and LEE features, are computed from the modes obtained after signal decomposition. Figure 10 and Figure 11 show different boxplots of mentioned entropy features obtained from EEG and ECG, respectively. The plots give information related to feature value distribution among different classes. Due to difference in the interquartile distance among different feature values good classification accuracy is obtained. Then, the extracted features are fed to two different classifiers: SVM and KNN. The above results show that the KNN classifier is found to have more accuracy than the SVM classifier. Further, the multi-model scenario is found to be more beneficial to reliably identify the emotional states of the subject. A statistical significance test was performed to show that the accuracy improvement in the multimodal emotion recognition model is significant [56]. An analysis of variance (ANOVA) test was performed on the 5-fold accuracies of different modalities, such as EEG, ECG, and multimodal. Figure 12 shows the accuracies of the high and low states of arousal, dominance, and valance using boxplots. The p-values for the statistical test for arousal, dominance, and valance are $1.33\times 10^{-18}$, $2.82\times 10^{-21}$, and $1.50\times 10^{-17}$, respectively, which indicates a significant improvement in accuracies for the multimodal emotion detection model. The proposed methodology is compared with the various existing approaches to identify the emotional states. The proposed method found to have superior performance as highlighted in Table 4 using bold fonts. In [57], various features related to EEG and ECG signals, such as power spectral density (PSD), HRV, entropy, EEG-topographical image-based, and ECG spectrogram image-based DL features, are calculated. Topic et al. [58] used EEG signals for emotion detection and has derived holographic features and for maximizing model accuracy, only relevant channels are selected. In [59], using data preprocessing, frame level features are derived from the EEG signal, from which, effective features are extracted using a teacher–student framework. In [60], the deep canonical correlation analysis (DCCA) method is proposed for emotion recognition. From the table, it may be noted that currently, various studies are being performed for emotion detection based on a DL-related framework [34,58,60,61,62] as it automatically finds the most significant features; however, it is difficult to understand or to find the reason behind the results obtained. Moreover, DL models are computationally complex and require a large amount of data to train [28]. Feature extraction in other existing state-of-the-art methods is a complex process and less accurate than the study presented here, which differentiates itself from the other existing studies as the results are encouraging. Thus, the proposed method has several advantages over the various listed DL-based feature extraction methods, such as being easy to comprehend, less complex, and more reliable. The disadvantage of the proposed multimodal approach is that its time complexity has increased compared to a single EEG-only modality. Still, the increased reliability makes it more suitable for feature extraction. As the results obtained using the proposed methodology have only been tested on a small size dataset, in the future, they can be tested on a larger database in order to verify their validity. Furthermore, in this study, a fixed 60 s duration physiological signals was used; this can also be made adaptive in order to select the time portion, which can make the methodology more efficient and adaptive.

## 5. Conclusions

This study presents an automated emotion identification by a multimodal approach using FB domain-based entropies. The publically available DREAMER data set was used to evaluate the performance of the proposed algorithm. Physiological signals obtained from the dataset, namely EEG and ECG, are decomposed using the FBSE-EWT into four modes. From these modes, several new FB-based entropy features, such as FB spectral-based SSE, LEE, and WE were computed. The dataset consisted of three-dimensional emotional space, i.e., arousal, valence, and dominance. Each of the dimensions are categorized into high and low classes based on the rating provided along with the dataset. The proposed method using the KNN classifier provides the highest accuracies of 97.84%, 97.91%, and 97.86% for arousal, valence, and dominance emotion classes, respectively. After comparing it with results obtained from current methods, significant improvement in the results is obtained. Moreover, the multimodal approach is found to be the most accurate in terms of identifying human emotional states.

## Figures and Tables

**Figure 1 entropy-24-01322-f001:**
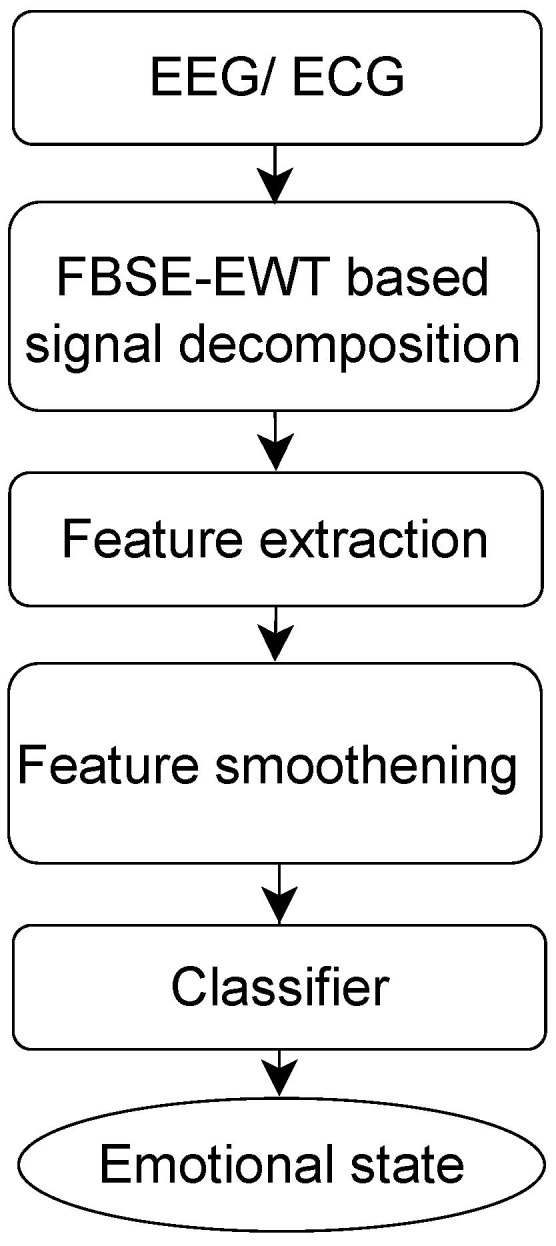
Block diagram for the proposed emotion detection methodology.

**Figure 2 entropy-24-01322-f002:**
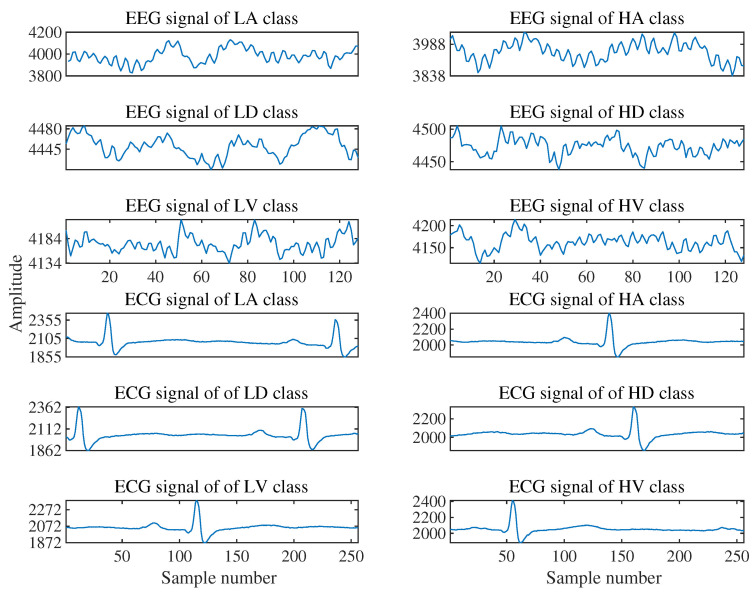
Epochs of raw EEG and ECG signals obtained from the DREAMER dataset.

**Figure 3 entropy-24-01322-f003:**
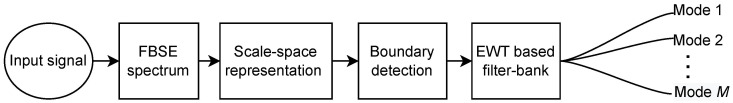
Flow diagram of FBSE-EWT based signal decomposition.

**Figure 4 entropy-24-01322-f004:**
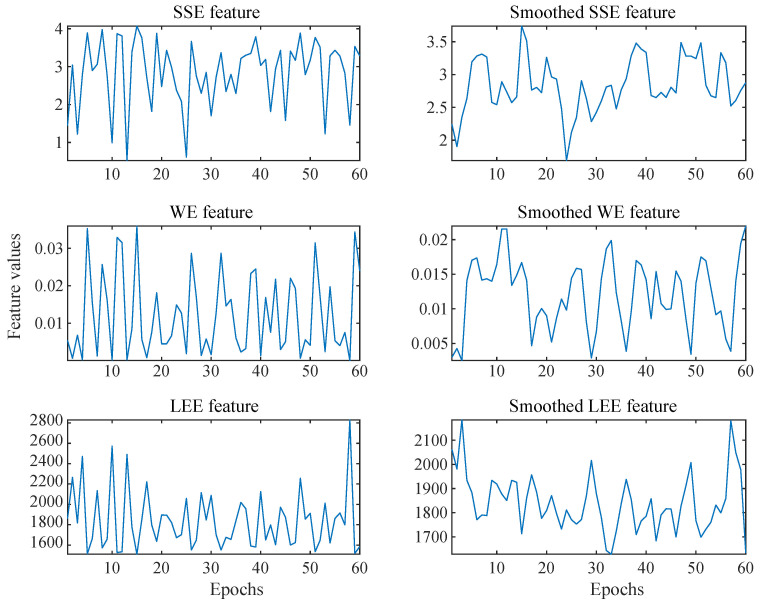
Feature values obtained from signal epochs is shown on the left and its smoothed version is shown on the right.

**Figure 5 entropy-24-01322-f005:**
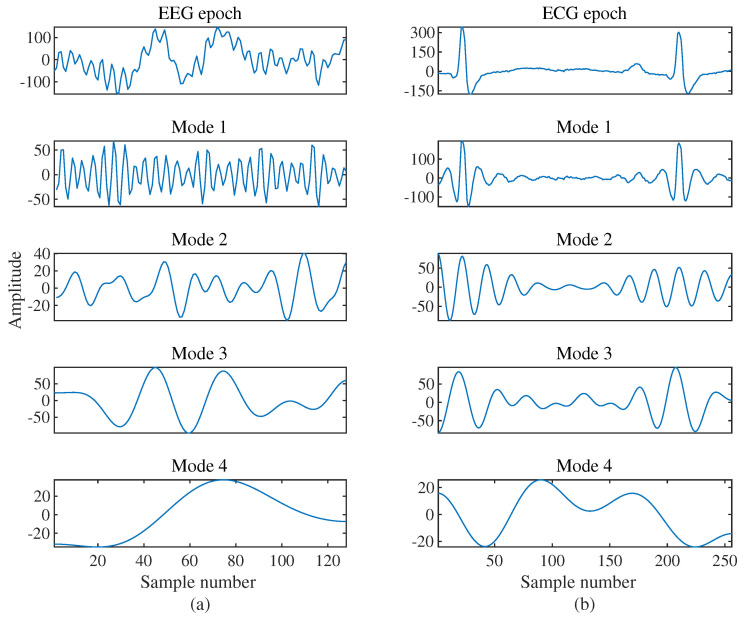
FBSE−EWT−based signal decomposition of (**a**) EEG and (**b**) ECG epochs into different modes.

**Figure 6 entropy-24-01322-f006:**
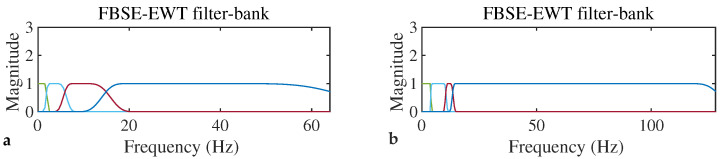
FBSE−EWT−based filter bank used for decomposing (**a**) EEG and (**b**) ECG epochs into different modes (filter corresponding to different modes are shown in different colors).

**Figure 7 entropy-24-01322-f007:**
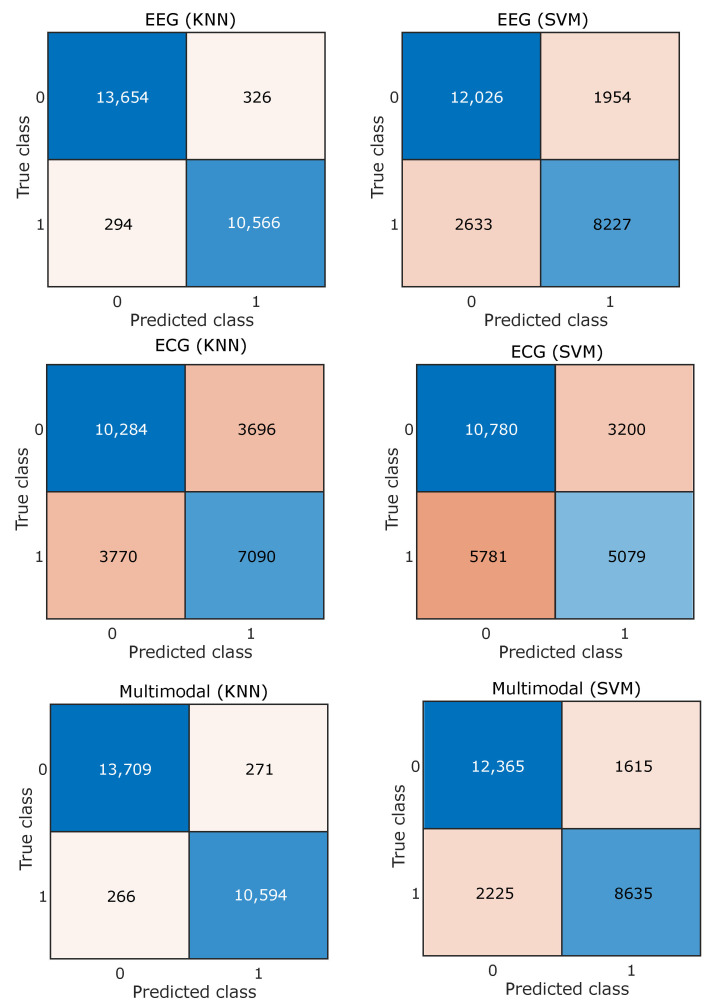
Confusion matrix for arousal emotion dimension.

**Figure 8 entropy-24-01322-f008:**
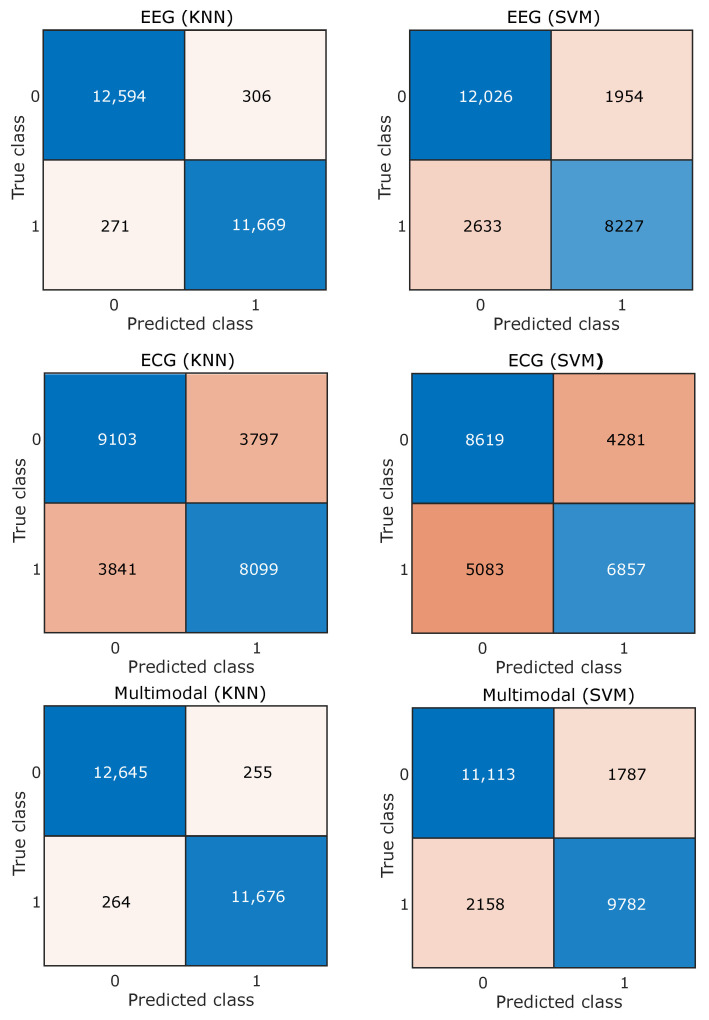
Confusion matrix for dominance emotion dimension.

**Figure 9 entropy-24-01322-f009:**
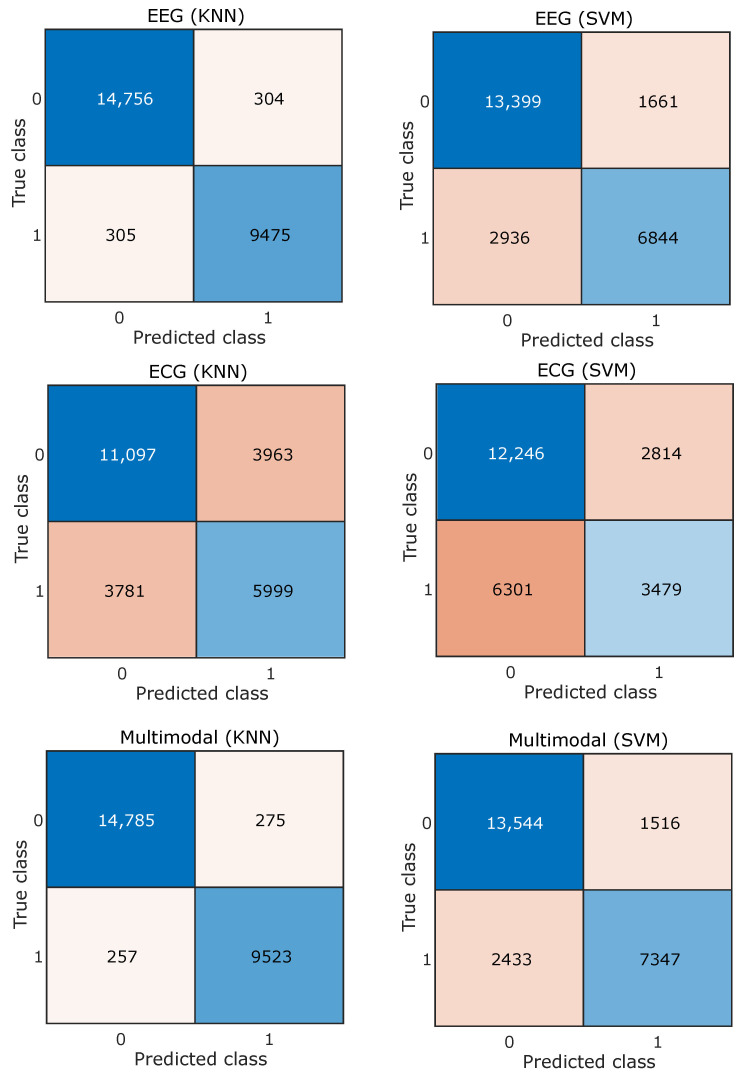
Confusion matrix for valence emotion dimension.

**Figure 10 entropy-24-01322-f010:**
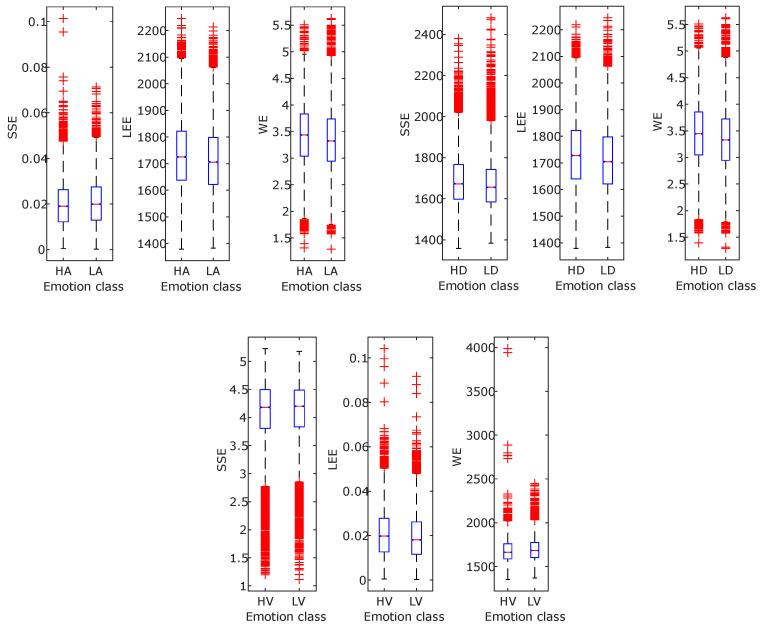
Box plots for the features calculated using EEG signals.

**Figure 11 entropy-24-01322-f011:**
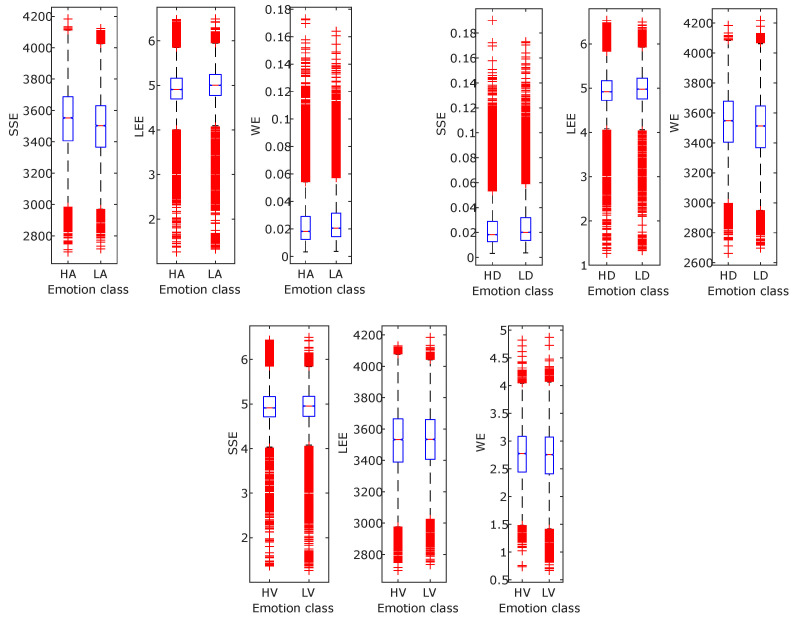
Box plots for the features calculated using ECG signals.

**Figure 12 entropy-24-01322-f012:**
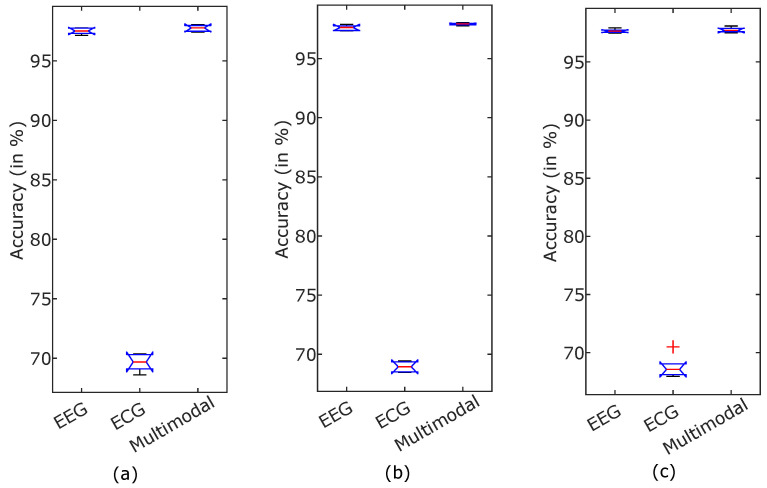
Box plots showing statistical significance of accuracies for different modalities (**a**) arousal, (**b**) dominance, and (**c**) valence.

**Table 1 entropy-24-01322-t001:** Performance with different modalities for arousal emotion dimension.

Modality	Model	Accuracy *	Sensitivity *	Specificity *	Precision *	F1 Score *
EEG	KNN (*k* = 1)	97.50	97.89	97.01	97.67	97.78
	SVM (Cubic)	81.53	82.04	80.81	86.02	83.98
ECG	KNN (*k* = 1)	69.94	73.17	65.73	73.56	73.37
	SVM (Cubic)	63.84	65.09	61.35	77.11	70.59
**Multimodal**	**KNN** (***k***** = 1**)	**97.84**	**98.10**	**97.51**	**98.06**	**98.08**
	SVM (Cubic)	84.54	84.75	84.24	88.45	86.56

* in percent.

**Table 2 entropy-24-01322-t002:** Performance with different modalities for dominance emotion dimension.

Modality	Model	Accuracy *	Sensitivity *	Specificity *	Precision *	F1 Score *
EEG	KNN (*k* = 1)	97.68	97.89	97.44	97.63	97.76
	SVM (Cubic)	81.53	82.04	80.81	86.02	83.98
ECG	KNN (*k* = 1)	69.25	70.33	68.08	70.57	70.45
	SVM (Cubic)	62.30	62.90	61.56	66.81	64.80
**Multimodal**	**KNN** (***k***** = 1**)	**97.91**	**97.95**	**97.86**	**98.02**	**97.99**
	SVM (Cubic)	84.12	83.74	84.55	86.15	84.93

* in percent.

**Table 3 entropy-24-01322-t003:** Performance with different modalities for valence emotion dimension.

Modality	Model	Accuracy *	Sensitivity *	Specificity *	Precision *	F1 Score *
EEG	KNN (*k* = 1)	97.55	97.97	96.89	97.98	97.98
	SVM (Cubic)	81.49	82.03	80.47	88.97	85.36
ECG	KNN (*k* = 1)	68.82	74.59	60.22	73.69	74.13
	SVM (Cubic)	63.31	66.03	55.28	81.31	72.88
**Multimodal**	**KNN** (***k***** = 1**)	**97.86**	**98.29**	**97.19**	**98.17**	**98.23**
	SVM (Cubic)	84.10	84.77	82.90	89.93	87.28

* in percent.

**Table 4 entropy-24-01322-t004:** Comparison of existing methodologies applied on DREAMER dataset with the proposed emotion recognition method.

Authors (Year)	Methodology	Modality	Accuracy (%)		
			LA-HA	LD-HD	LV-HV
Katsigiannis et al. [32] (2018)	PSD features and SVM classifier	EEG & ECG	62.32	61.84	62.49
Song et al. [62] (2018)	DGCNN	EEG	84.54	85.02	86.23
Zhang et al. [34] (2019)	PSD features and GCB-net classifier	EEG	89.32	89.20	86.99
Bhattacharyya et al. [28] (2020)	MFBSE-EWT based entropy features and ARF classifier	EEG	85.4	86.2	84.5
Cui et al. [61] (2020)	RACNN	EEG	97.01	-	95.55
Kamble et al. [63] (2021)	DWT, EMD-based features, and CML and EML-based classifier	EEG	93.79	-	94.5
Li et al. [64] (2021)	3DFR-DFCN	EEG	75.97	85.14	82.68
Siddharth et al. [57] (2022)	PSD, HRV, entropy, DL based feature, and LSTM based classifier	EEG & ECG	79.95	-	79.95
Topic et al. [58] (2022)	Holographic features and CNN	EEG	92.92	92.97	90.76
Gu et al. [59] (2022)	FLTSDP	EEG	90.61	91.00	91.54
Liu et al. [60] (2022)	DCCA	EEG & ECG	89.00	90.7	90.6
**Proposed work**	**FBSE-EWT-based entropy features** **and KNN classifier**	**EEG** **&** **ECG**	**97.84**	**97.91**	**97.86**

PSD = power spectral density, SVM = support-vector machines, DGCNN = dynamic graph convolutional neural network, GCB-net = graph convolutional broad network, MFBSE-EWT = multivariate FBSE-EWT, ARF = adaptive random forest, RACNN = region-based convolutional neural network, DWT = discrete wavelet transform, EMD = empirical mode decomposition, CML = conventional machine learning, 3DFR-DFCN = 3-D feature representation and dilated fully convolutional network, HRV = heart rate variability, DL = deep learning, LSTM = long short term memory, CNN = convolutional neural network, FLTSDP = frame level teacher-student framework with data privacy, DCCA = deep canonical correlation analysis.

## Data Availability

The EEG and ECG data are provided by the Stamos Katsigiannis collected at University of the West of Scotland.

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
