# Peer review of "Automated Emotion Identification Using Fourier–Bessel Domain-Based Entropies"

_entropy, 2022, doi:10.3390/e24101322_

Round 1

Reviewer 1 Report

1-your data is public or private?

2-improve discussion section

3-what is advantage and disadvantage of your method?

4-make sure your figures do not need any reference

5-compare your results with more papers in this field

Author Response

  1. your data is public or private?

Response:

Thank you for this query. We have used a publicly available DREAMER dataset, as mentioned in section 2.1 of the article.

  1. improve discussion section

Response:

Thank you for your suggestion. As per your suggestion, we have included more details related to various existing state-of-the-art methods and compared them with our method.

  1. what is advantage and disadvantage of your method?

Response:

Thank you for pointing out the issue; it will increase the article's readability. We have added more descriptions in the discussion section itself. In this revised version, we have described various advantages of the proposed method and pointed out some limitations.

  1. make sure your figures do not need any reference

Response:

Thank you for putting forward this question. We confirm that all the figures presented are original and are not taken from anywhere else.  

  1. compare your results with more papers in this field

Response:

Thank you for your comments which will help the reader to gain confidence in our work further. Table 4 is updated as per your suggestion.

Reviewer 2 Report

The authors presented FBSE-based empirical wavelet transform for EEG/ECG based emotion classification. The paper is nicely written and easy to follow.

I have the following comments:

1. The authors need to include other works on DREAMER for emotion classification in Table 4 for the comparison with state-of-the-art methods. 

2. Authors used 5-fold cross-validation. How about subject independent test to avoid subject dependency? The presented 5-fold CV approach will have samples from the same subject in both test and training data.

3. Fig. 6: what are the frequency ranges of those decomposed modes (for both the EEG and ECGs)? This might be interesting for the readers. Authors can mention this in the text/figure. Why the EEG and ECGs are decomposed into four modes? Is there any physiologic reasoning for selecting the number 4?

4. "Rating ⩽ 3 is consider low and all above it is considered as high"-is this the same strategy followed by the other works? Authors can include the related references here.

Author Response

The authors presented FBSE-based empirical wavelet transform for EEG/ECG based emotion classification. The paper is nicely written and easy to follow.

Response:

Thank you for your encouraging words. Additionally, we would like to thank you for your valuable suggestions, which have helped us avoid flaws. We have carefully considered each of the comments on this revision and tried our best to answer those.

  1. The authors need to include other works on DREAMER for emotion classification in Table 4 for the

comparison with state-of-the-art methods.

Response:

Thank you for your valuable suggestions. We have included a few more existing methods evaluated using the DREAMER dataset for a more explicit comparison with our method in Table 4.

  1. Authors used 5-fold cross-validation. How about subject independent test to avoid subject dependency? The presented 5-fold CV approach will have samples from the same subject in both test and training data.

Response:

Thank you for pointing out the issue. We have missed mentioning that subject independent test has been undertaken, and therefore the data used for training and testing is chosen completely randomly. The result section is updated accordingly. 

  1. 6: what are the frequency ranges of those decomposed modes (for both the EEG and ECGs)?

This might be interesting for the readers. Authors can mention this in the text/figure. Why the EEG

and ECGs are decomposed into four modes? Is there any physiologic reasoning for selecting the

number 4?

Response:

Thank you for raising the queries. As per your suggestion, we have shown the filter bank in Fig. 7, which has been used for decomposing EEG and ECG signals (in Fig. 6) into different modes. Also, from our preliminary experiment, we observed that maximum epochs gave a minimum of four modes, so we set the number of decomposed modes to four.

  1. "Rating ⩽ 3 is consider low and all above it is considered as high"-is this the same strategy

followed by the other works? Authors can include the related references here.

Response:

Thank you for this query. Yes, as per literature survey, other authors' work has also considered the same threshold for their work, and the reference is also cited in the result section of the article.

Reviewer 3 Report

The paper reports about the implementation of an automated method to classify emotions based on Fourier-Bessel domains entropies. The paper is well written, and the results seem ok. In my opinion some concerns need to be addressed before publication.

1)     The paper is well structured. However, in my opinion figures 8-10 and tables 1-4 reports results, thus maybe the Authors could consider moving them in the results section

2) Concerning the tables, it could be useful for the reader if the best performances are highlighted.

 3)     A concern is about the 5-folds cross validation. Each participant watched 18 videos, thus several observations came from each subject. Were the observations from each participant placed in the same fold? Because, whether samples from the same participant are present in different folds, it could introduce some overfitting effects. Please specify this aspect in the manuscript.

 4)     Please better specify in the manuscript how the classes were defined, and how many classes were defined. Moreover, specify whether the classes were balanced.

 5)     In the results section, it could be interesting to compare statistically the accuracies obtained with the different classifiers and modalities, in order to demonstrate whether the multimodal approach is significantly superior to the unimodal approach.

 6)     Finally, in my opinion it could be interesting to describe, in the introduction, the literature regarding the emotion recognition from  several physiological signals. For instance, the Authors could refer to:

 Filippini, C., Di Crosta, A., Palumbo, R., Perpetuini, D., Cardone, D., Ceccato, I., ... & Merla, A. (2022). Automated Affective Computing Based on Bio-Signals Analysis and Deep Learning Approach. Sensors22(5), 1789.

 Kipli, K., Latip, A. A. A., Lias, K., Bateni, N., Yusoff, S. M., Tajudin, N. M. A., ... & Mahmud, M. (2022). GSR Signals Features Extraction for Emotion Recognition. In Proceedings of Trends in Electronics and Health Informatics (pp. 329-338). Springer, Singapore.

 Dutta, S., Mishra, B. K., Mitra, A., & Chakraborty, A. (2022). An Analysis of Emotion Recognition Based on GSR Signal. ECS Transactions107(1), 12535.

Author Response

The paper reports about the implementation of an automated method to classify emotions based on Fourier-Bessel domains entropies. The paper is well written, and the results seem ok. In my opinion some concerns need to be addressed before publication.

Response:

Thank you for reading our article and giving your valuable suggestions to us. We have carefully considered each of the comments in this revision and tried our best to answer them.

  • The paper is well structured. However, in my opinion figures 8-10 and tables 1-4 reports results, thus maybe the Authors could consider moving them in the results section

Response:

Thank you for your valuable suggestion. As per your suggestion, we have moved the figures and table to the appropriate place, but some tables and figures are coming in the discussion section due to automatic placement by Latex.

  • Concerning the tables, it could be useful for the reader if the best performances are highlighted.

Response:

Thank you for your comments which helped in making our tables more presentable to the reader. As per your suggestion, we have highlighted the best performance results using bold fonts.

  • A concern is about the 5-folds cross-validation. Each participant watched 18 videos, thus several observations came from each subject. Were the observations from each participant placed in the same fold? Because, whether samples from the same participant are present in different folds, it could introduce some overfitting effects. Please specify this aspect in the manuscript.

Response:

Thank you for mentioning the above concern. We have done cross-validation by randomly choosing training and testing data as we have done a subject independent test over the given data. The same has been updated in the result section of the revised article.

  • Please better specify in the manuscript how the classes were defined, and how many classes were defined. Moreover, specify whether the classes were balanced.

Response:

Thank you for the suggestion, which helps in avoiding confusion. We have considered the binary classification of the three-emotional dimension, i.e., arousal, dominance, and valence, into a low and high class of each emotion dimension. The available DREAMER dataset is not balanced for all participants. The same is now mentioned in the result section of the article with proper referencing.

  • In the results section, it could be interesting to compare statistically the accuracies obtained with the different classifiers and modalities, in order to demonstrate whether the multimodal approach is significantly superior to the unimodal approach.

Response:

Thank you for your valuable suggestions. We have performed a statistical significance test to show the accuracy improvement in the multimodal emotion recognition model is significant. ANOVA test is performed on the 5-fold accuracies of different modalities like EEG, ECG, and multimodal. Fig. 12 shows the accuracies of the high and low states of arousal, dominance, and valance using boxplots. The p-values for the statistical test for arousal, dominance, and valance are 1.33*10-18, 2.82*10-21, and 1.50*10-17, respectively, which indicates a significant improvement in accuracies for the multimodal emotion detection model.

  • Finally, in my opinion it could be interesting to describe, in the introduction, the literature regarding the emotion recognition from  several physiological signals. For instance, the Authors could refer to:

 Filippini, C., Di Crosta, A., Palumbo, R., Perpetuini, D., Cardone, D., Ceccato, I., ... & Merla, A. (2022). Automated Affective Computing Based on Bio-Signals Analysis and Deep Learning Approach. Sensors, 22(5), 1789.

 Kipli, K., Latip, A. A. A., Lias, K., Bateni, N., Yusoff, S. M., Tajudin, N. M. A., ... & Mahmud, M. (2022). GSR Signals Features Extraction for Emotion Recognition. In Proceedings of Trends in Electronics and Health Informatics (pp. 329-338). Springer, Singapore. 

 Dutta, S., Mishra, B. K., Mitra, A., & Chakraborty, A. (2022). An Analysis of Emotion Recognition

Based on GSR Signal. ECS Transactions, 107(1), 12535.

         Response:

Thank you for your comments which will help in extending the literature related to the emotion recognition system in our article. We have included work related to other physiological signals in our introduction section.

Reviewer 4 Report

This paper proposes novel entropy-based features in the Fourier-Bessel domain instead of the Fourier domain, where frequency resolution is twice as compared to the latter.

The paper is well organized and readable. I have only one significant suggestion, described below, to be considered to improve the paper.

The paper presents that the multimodal approach is found to be the most accurate in terms of identifying human emotional states. I have no problem with this finding. However, the paper lacks comparative experiments with the other latest methods in this field. The papers presented in the journals with the impact factor should also include comparison experiments by other similar methods.

I recommend that the paper should be accepted with major revision.

Author Response

This paper proposes novel entropy-based features in the Fourier-Bessel domain instead of the Fourier domain, where frequency resolution is twice as compared to the latter.

The paper is well organized and readable. I have only one significant suggestion, described below, to be considered to improve the paper.

The paper presents that the multimodal approach is found to be the most accurate in terms of identifying human emotional states. I have no problem with this finding. However, the paper lacks comparative experiments with the other latest methods in this field. The papers presented in the journals with the impact factor should also include comparison experiments by other similar methods.

I recommend that the paper should be accepted with major revision.

Response:

Thank you for your encouraging words. As per your comment, we have included more related work by updating Table 4, which comprises both unimodal and multimodal approaches for emotional state identification.

Round 2

Reviewer 2 Report

The authors have addressed my previous concerns

Reviewer 3 Report

The Authors addressed all my concerns and the paper is strongly improved and, in my opinion, it is suitable for publication in the present form.

Reviewer 4 Report

This paper proposes novel entropy-based features in the Fourier-Bessel domain instead of the Fourier domain, where frequency resolution is twice as compared to the latter.

The paper is well organized and readable. I have only one suggestion, described below, to be considered to improve the paper.

The authors presented the results of additional comparative experiments and responded to my recommendation. I would only suggest that the authors redesign Table 4 because it is also presented on the right margin of the paper.

I recommend that the paper should be accepted with minor revision.